# The English (H6R) Mutation of the Alzheimer’s Disease Amyloid-β Peptide Modulates Its Zinc-Induced Aggregation

**DOI:** 10.3390/biom10060961

**Published:** 2020-06-25

**Authors:** Sergey P. Radko, Svetlana A. Khmeleva, Dmitry N. Kaluzhny, Olga I. Kechko, Yana Y. Kiseleva, Sergey A. Kozin, Vladimir A. Mitkevich, Alexander A. Makarov

**Affiliations:** 1Engelhardt Institute of Molecular Biology, Russian Academy of Sciences, 119991 Moscow, Russia; uzhny@mail.ru (D.N.K.); olga.kechko@gmail.com (O.I.K.); kozinsa@gmail.com (S.A.K.); mitkevich@gmail.com (V.A.M.); aamakarov@eimb.ru (A.A.M.); 2Institute of Biomedical Chemistry, 119121 Moscow, Russia; diny1204@yandex.ru; 3Russian Scientific Center of Roentgenoradiology, 117485 Moscow, Russia; yana.kiseleva@gmail.com

**Keywords:** amyloid-β, English mutation, zinc, aggregation

## Abstract

The coordination of zinc ions by histidine residues of amyloid-beta peptide (Aβ) plays a critical role in the zinc-induced Aβ aggregation implicated in Alzheimer’s disease (AD) pathogenesis. The histidine to arginine substitution at position 6 of the Aβ sequence (H6R, English mutation) leads to an early onset of AD. Herein, we studied the effects of zinc ions on the aggregation of the Aβ42 peptide and its isoform carrying the H6R mutation (H6R-Aβ42) by circular dichroism spectroscopy, dynamic light scattering, turbidimetric and sedimentation methods, and bis-ANS and thioflavin T fluorescence assays. Zinc ions triggered the occurrence of amorphous aggregates for both Aβ42 and H6R-Aβ42 peptides but with distinct optical properties. The structural difference of the formed Aβ42 and H6R-Aβ42 zinc-induced amorphous aggregates was also supported by the results of the bis-ANS assay. Moreover, while the Aβ42 peptide demonstrated an increase in the random coil and β-sheet content upon complexing with zinc ions, the H6R-Aβ42 peptide showed no appreciable structural changes under the same conditions. These observations were ascribed to the impact of H6R mutation on a mode of zinc/peptide binding. The presented findings further advance the understanding of the pathological role of the H6R mutation and the role of H6 residue in the zinc-induced Aβ aggregation.

## 1. Introduction

The aggregation of amyloid-β peptide (Aβ) is considered as a crucial event in pathogenesis of Alzheimer’s disease (AD)—a devastating neurodegenerative disorder which affects tens of millions of people throughout the world [1]. Though molecular mechanisms underlying Aβ aggregation were a subject of intensive in vitro and in vivo studies for about three decades, they are still incompletely understood [2,3]. Among factors which can potentially contribute to the in vivo Aβ aggregation, zinc ions (Zn^2+^) have attracted a constant interest [4,5] ever since they were found to trigger rapid Aβ aggregation in vitro [6]. It is worth noting that amyloid plaques (the deposits in brain parenchyma, primarily composed of 40- and 42-amino-acid-long Aβ peptides—Aβ40 and Aβ42, correspondingly), which are a hallmark of AD [7], are abnormally loaded with zinc [8,9]. Being an important neuromodulator, zinc, when released from synaptic vesicles during neuronal excitation, can be transiently present in a synaptic cleft at considerably high concentrations—up to 300 µM [10]. Interestingly, the transgenic AD model mice with a knocked-out gene coding for the specific zinc transporter Zn-T3 (responsible for pumping zinc into synaptic vesicles) do not develop amyloid plaques [11]. The plausible role of zinc in AD pathogenesis as a causative agent alongside that of another vital metal such as copper has inspired a development of the approach known as “AD chelation therapy” [12], as a part of efforts to find a cure for the disease.

It is known that Zn^2+^ triggers the formation of Aβ aggregates which lack the defined cross-β-sheet organization typical for Aβ fibrils and are often referred to as “amorphous” aggregates [10]. The Zn^2+^-induced Aβ aggregation is undoubtedly initiated by a formation of Zn^2+^-Aβ complexes: Aβ peptides bind zinc via the Zn^2+^ coordination involving histidine residues H6, H13, and H14 of the peptide’s N-terminal region consisting of amino acid residues 1–16 and referred to as a metal-binding domain (MBD) [10,13]. Due to the importance of histidine residues for the Zn^2+^ coordination, their role in the Zn^2+^-dependent Aβ aggregation has been probed in a number of studies, using various substitutions for histidine residues in truncated and full-length Aβ peptides (Aβ28, Aβ40, and Aβ42) [14,15,16]. The substitution of alanine residue for either H13 or H14, but not H6, was demonstrated to suppress the effect of Zn^2+^ on the fibrillar aggregation of these peptides [15,16]. The double substitution—R5 and H6 with alanine residues—was also found to have no impact on the Zn^2+^-triggered “coil-to-β-sheet” conformational transition in Aβ28 and Aβ40 peptides [15]. However, as we have earlier shown, the substitution of H6 with arginine residue (H6R, known as the “English mutation” associated with the early onset of AD [17]), while not preventing the Zn^2+^-induced Aβ42 aggregation, resulted nonetheless in some reduction of the number of sedimentation-prone Zn^2+^-induced aggregates (at Zn^2+^/peptide ratios of 1 to 3), compared to the wild-type Aβ42 [18]. Aβ42 peptides bearing the H6R mutation (H6R-Aβ42) were also shown to produce Zn^2+^-induced aggregates of a smaller size at a Zn^2+^/peptide ratio below 1 [19]. Still, many aspects of how the H6R mutation can affect the Zn^2+^-induced aggregation of full-length Aβ peptides are unclear yet.

The aim of the present work was to get further insights into the effects of H6R substitution on the in vitro aggregation of the Aβ42 peptide, triggered by Zn^2+^. To achieve this aim, the Zn^2+^-induced aggregation of synthetic Aβ42 and H6R-Aβ42 peptides in the presence of various amount of Zn^2+^ was studied with circular dichroism spectroscopy (CD), dynamic light scattering (DLS), turbidimetric and sedimentation methods, and fluorescence assays utilizing bis-ANS and thioflavin T fluorescent dyes. The Aβ42 and H6R-Aβ42 peptides were found to attain different conformations in Zn^2+^-induced amorphous aggregates upon complexing with Zn^2+^ that leads to a formation of aggregates with distinct characteristics.

## 2. Materials and Methods 

### 2.1. Materials

Synthetic peptides Aβ42 (DAEFR**H**DSGYEVHHQKLVFFAEDVGSNKGAIIGLMVGGVVIA) and H6R-Aβ42 (purity > 95%) were purchased as a lyophilized solid from Biopeptide Co., LLC (San Diego, CA, USA). The Aβ42 sequence is presented in the parenthesis. The residue marked in bold is substituted in the H6R-Aβ42 isoform with the arginine residue. All chemicals used were obtained from Sigma-Aldrich (St. Louis, MO, USA) and were of an analytical grade or higher. Chloride of zinc served as a source of zinc ions. The Milli-Q quality water was used to prepare all solutions; Milli-Q water and stock buffer solutions were filtered through a 0.22 µm pore syringe filter (Merck, Kenilworth, NJ, USA, #SLGP033RB).

### 2.2. Peptide Solutions and Preparation of Zinc-Induced Aβ42 Aggregates

To prepare Aβ42 solutions, Aβ42 peptides were treated with hexafluoroisopropanol (HFIP), dried, and dissolved in 10 mM NaOH at a concentration of 0.5 mM. The Aβ42 solutions in 10 mM NaOH were adjusted with 100 mM HEPES-buffer (pH 5.0) to pH 6.8 and subjected to centrifugation (30 min, 16,000× *g*, 4 °C) to remove insoluble peptide aggregates. Peptides in the supernatant were quantified spectrophotometrically (based on the molar extinction coefficient ε_280_ = 1490 M^−1^ × cm^−1^ [20]) and diluted with appropriate buffers to provide Aβ42 solutions of desired concentrations in the buffer containing 10 mM HEPES (pH 6.8) and 50 mM NaCl (further referred to as ‘buffer H’). The peptide solutions were kept on ice until further use. H6R-Aβ42 peptide solutions were prepared in an identical manner.

The Zn^2+^-induced Aβ aggregation was triggered by mixing aliquots of 40-µM peptide solutions with buffer H supplemented with ZnCl_2_ at various concentrations. The mixtures were incubated under quiescent conditions at room temperature for 30 min. For bis-ANS (4,4′-dianilino-1,1′-binaphthyl-5,5′-disulfonic acid)-based fluorescence measurements, an aliquot of the stock bis-ANS solution (700 µM in buffer H) was added to the peptide solution at the peptide/bis-ANS molar ratio of 10:1 prior to inducing peptide aggregation. The final peptide concentration in all cases was 25 µM.

### 2.3. Circular Dichroism Spectroscopy

CD spectra of Aβ42 and H2R-Aβ42 peptides were acquired using a J-715 spectropolarimeter (JASCO, Easton, MD, USA). A 25-µM solution of Aβ peptides either in a monomeric or aggregated state in buffer H was placed into a quartz cell with a path length of 1 mm, and CD spectra were collected from 195 to 260 nm with a 1 nm interval at 25 °C. Every CD spectrum was an average of three separate measurements. The Aβ aggregation was induced 20 min prior to a measurement by adding Zn^2+^ to the Aβ solutions at Zn^2+^/Aβ molar ratios of 2 and 4. The analysis of the peptides’ secondary structure was carried out using software package CDNN [21,22].

### 2.4. Turbidimetry and Dynamic Light Scattering

The Zn^2+^-induced aggregation of Aβ42 and H6R-Aβ42 peptides was evaluated by measuring optical density of Aβ preparations at 405 nm (OD_405_) on an Agilent 8453E spectrophotometer (Agilent Technologies, Santa Clara, CA, USA). The OD_405_ values measured in the absence of Zn^2+^ were taken as the initial (zero time) values.

DLS measurements were carried out on a Zetasizer Nano ZS apparatus (Malvern Instruments Ltd., Malvern, UK) at 25 °C as described elsewhere [18,23]. The apparatus is able to measure particles sizes in the range of 0.6 nm to 10 µm. The characteristic size of Aβ aggregates was expressed in terms of an average “diameter”, since instrument software approximates the heterogeneous population of Aβ aggregates by a population of spherical particles with the identical distribution of a diffusion coefficient. The number particle distribution was used by the instrument software to calculate average particle diameters.

### 2.5. Sedimentation Assay

Aβ preparations with and without added Zn^2+^ were subjected to centrifugation (16,000× *g*, 10 min, 20 °C). Supernatants were sampled and peptide concentrations were determined with a commercial BCA assay (Thermo Fisher Scientific, Waltham, MA, USA, #23235). The relative amount of peptides in the supernatant was calculated as *C/C*_0_, where *C* and *C*_0_ are peptide concentrations in the supernatant in the presence and absence of Zn^2+^, respectively.

### 2.6. Fluorimetry

Fluorescence measurements were carried out on an Infinite M200 PRO microplate reader (TECAN, Männedorf, Switzerland) using Corning 96-well microplates. For bis-ANS fluorescence measurements, aliquots of bis-ANS containing Aβ preparations were placed into wells in triplicates and aggregation was initiated by the addition of Zn^2+^-containing buffer H. The final volume of each sample was 100 µL. After the 30-min incubation, fluorescence was recorded using 400 and 500 nm wavelengths for excitation and emission, respectively. The fluorescence was corrected by subtracting values of fluorescence in wells with merely bis-ANS in buffer H to provide the “pure” bis-ANS fluorescence of bis-ANS/Aβ complexes, *F*. To perform thioflavin T (4-(3,6-dimethyl-1,3-benzothiazol-3-ium-2-yl)-N,N-dimethylaniline chloride, ThT)-based fluorescence measurements, Aβ preparations were mixed with aliquots of the ThT solution in buffer H so as to provide the final Aβ and ThT concentrations of 25 µM each. The aliquots of Aβ/ThT mixtures were placed in wells in triplicates and the Zn^2+^-induced aggregation was initialed as above. The ThT fluorescence was measured as described above, except for excitation and emission wavelengths which were correspondingly set at 450 and 482 nm.

## 3. Results

### 3.1. The Zn^2+^-Induced Aggregation of Aβ42 and H6R-Aβ42 Peptides Measured with Turbidity, DLS, and Sedimentation Methods

The addition of Zn^2+^ to solutions of Aβ42 and its mutated isoform, H6R-Aβ42, triggered a rapid peptide aggregation manifested by a rise in solution turbidity (Figure 1). Turbidity values appeared to approach a plateau by 30 min of incubation for all Zn^2+^/peptide molar ratios tested. The OD_405_ values measured at 30-th min of incubation were flattened out at the Zn^2+^/peptide ratios of 2 and above, showing considerably higher turbidity for the mutant peptide, compared to the intact (the insert in Figure 1A). The sizes of Aβ42 and H6R-Aβ42 aggregates measured by DLS after 30-min incubation were indistinguishable within the experimental scatter and equal to 1.5–2 µm in diameter for both peptides at Zn^2+^/peptide ratios of 1 to 4 (Figure 2). In the absence of Zn^2+^, the species with a characteristic size of about 14–17 nm were detected by DLS in the Aβ42 and H6R-Aβ42 solutions within the 30-min incubation interval that may be attributed to the presence of low molecular weight Aβ42 oligomers [24]. Since at Zn^2+^/peptide molar ratios of 2 to 4, the size of Aβ aggregates was similar for both isoforms (Figure 2) while turbidity values significantly differed (Figure 1), we tested by the sedimentation method whether the number of peptides involved in the formation of Zn^2+^-induced aggregates varied for Aβ42 and H6R-Aβ42 isoforms. At the Zn^2+^/peptide molar ratio of 4, the fraction of Aβ42 and H6R-Aβ42 peptides included into sedimentation-prone Zn^2+^-induced aggregates was found to equal (0.91±0.04) and (0.86±0.05), respectively. Thus, practically all peptides were in an aggregated state for both isoforms at that Zn^2+^/peptide ratio. Consequently, at least at the highest Zn^2+^/peptide ratio used, the observed differences in turbidity of Aβ42 and H6R-Aβ42 preparations (Figure 1) could be accounted for by neither the aggregate size (which is similar, Figure 2) nor the number of aggregated peptides.

### 3.2. Effect of the H6R Mutation on Zn^2+^-Induced Conformational Changes in Aβ Peptides

In the absence of Zn^2+^, CD spectra for Aβ42 and H6R-Aβ42 peptides were found to slightly differ (Figure 3A,B). Deconvolution of the CD spectra revealed that the native Aβ42 peptide possessed predominantly an α-helix/random coil secondary structure (Figure 3C). Such structural organization was observed for HFIP-treated Aβ peptides dissolved in aqueous solutions and ascribed to their monomeric state [25,26]. For the mutant peptide, H6R-Aβ42, the random coil organization dominated, and an increase in β-sheet structure, compared with the intact peptide, was noticeable (Figure 3D). This observation agreed with the molecular dynamic simulation results on structural characteristics of Aβ42 and H6R-Aβ42 peptides [27]. A higher β-structure content in Aβ peptides was suggested to promote their self-aggregation [28,29] and may have been responsible for the known higher self-aggregation propensity of the H6R-Aβ mutants compared with the intact Aβ peptides [30,31]. Upon addition of Zn^2+^, the CD spectrum changed for the Aβ42 peptide, while no changes were observed practically for H6R-Aβ42 (Figure 3A,B). Consequently, for H6R-Aβ42, the Zn^2+^-induced aggregation was not accompanied by sufficient alterations in peptide structure (Figure 3C). At the same time, the intact peptide underwent a structural reorganization manifested by a steady increase in random coil and β-sheet content with Zn^2+^ concentration (Figure 3D). The Zn^2+^-triggered increase in β-sheet content was previously reported for the Aβ42 peptide [32,33]. Interestingly, another divalent metal, copper, also known to trigger the rapid Aβ aggregation, was shown to induce a similar transition from the α-helix to random coil and β-sheet conformations in the Aβ40 peptide [34]. Thus, the results of CD analysis indicate that Aβ42 and H6R-Aβ42 peptides attain different conformations in Zn^2+^-induced amorphous aggregates upon complexing with Zn^2+^. That, in turn, may lead to a distinct structural organization of these aggregates per se.

### 3.3. Zn^2+^-Induced Aggregation of Aβ Isoforms Tested with ThT and Bis-ANS Fluorescence Assays

Since the Aβ42 peptide demonstrated an increase in the β-structure content upon the addition of Zn^2+^, we tested Aβ42 and H6R-Aβ42 preparations with ThT (whose fluorescence is known to greatly increase upon the binding to fibrillar amyloid aggregates [35]) in order to ensure that no fibrillar Aβ aggregates were formed under our experimental conditions. Indeed, no appreciable differences in ThT fluorescence were observed in both the absence and presence of Zn^2+^ (Zn^2+^/peptide ratios of 1 to 4) after the 30-min incubation period for both isoforms (data not shown). Hence, one may conclude that merely amorphous Aβ aggregates were formed under the experimental conditions used.

As an addition to the CD analysis, we qualitatively probed the structure of Zn^2+^-induced Aβ42 and H6R-Aβ42 aggregates with bis-ANS—a sulfonated naphthalene derivative whose fluorescence noticeably increases in a nonpolar surrounding [36]. The dye is known to bind to proteins mostly via hydrophobic interactions and is able to report the exposure of hydrophobic clusters on protein surface as a result of a specific conformational reorganization or denaturation [36]. Bis-ANS has been employed for monitoring the Zn^2+^-induced oligomerization and aggregation of Aβ peptides [37,38,39]. Presumably, the enhancement of the dye’s fluorescence during the aggregation process is related to rearrangements in the Aβ conformation, triggered by the Zn^2+^ binding and the inclusion of Aβ into oligomers and amorphous aggregates [37,38].

The fluorescence of bis-ANS/Aβ complexes, *F*, increased with the Zn^2+^ load, reaching a plateau at the Zn^2+^/peptide ratio of 2 for both Aβ42 and H6R-Aβ42 isoforms (Figure 4). The *F* values were higher for Aβ42 at the plateau than for H6R-Aβ42, and the differences were statistically significant (*p* < 0.05, the two-tailed Student’s *t*-test). Below the Zn^2+^/peptide ratio of 2, no statistically significant differences in *F* values between peptides were observed (*p* > 0.05). In the absence of Zn^2+^, *F* values showed no appreciable changes within the 30-min incubation period for both isoforms. Hence, the results of bis-ANS fluorescence analysis support the assumption that Zn^2+^-induced Aβ42 and H6R-Aβ42 aggregates are structurally different.

## 4. Discussion

The pathogenic mutations in the Aβ sequence, associated with the early onset of AD, are responsible for only a small percentage of all AD cases. Nonetheless, these mutations, alongside with artificial amino acid substitutions, attract a constant interest since they demonstrate a variety of Aβ aggregation pathways in vitro [40,41], thus allowing a better understanding of molecular mechanisms underlying the pathological Aβ aggregation [42]. It is well established that the English familiar mutation, H6R, promotes the self-oligomerization and fibrillar aggregation of the full-length Aβ peptides [30,31]. Our study elucidates the pathological role of H6R mutation in the Zn^2+^-triggered aggregation of these peptides.

The present experimental results demonstrate that the H6R mutation in Aβ42 peptides, while not suppressing the Zn^2+^-triggered aggregation under molar excess of Zn^2+^, as the H13A and H14A substitutions did [16], can still markedly alter the character of Zn^2+^-induced amorphous aggregates. The approximately twice higher turbidity of H6R-Aβ42 preparations at the Zn^2+^/peptide ratios of 2 and above, compared with that for Aβ42 preparations (Figure 1), altogether with the similar aggregate characteristic size (Figure 2) and degree of aggregation, suggests that optical properties (most probably, the refractive index) of formed amorphous aggregates as scattering centers are quite different. Since the aggregates are amorphous and, therefore, lack a clearly defined structural organization, this difference may reflect a variation in a density of peptide random packing. Apparently, the dissimilar structural organization of individual Aβ peptides in these aggregates, revealed by CD spectroscopy (Figure 3), determines the dissimilar peptide packing. It seems fair to suggest that different modes of Zn^2+^/Aβ complexing govern the formation of distinct Zn^2+^-induced aggregates in the case of Aβ42 or H6R-Aβ42 peptides.

The effects of H6R substitution on the Zn^2+^/Aβ complexing were thoroughly studied using the Aβ16 peptide as a convenient model of Aβ MBD [43,44,45]. In the intact Aβ16, a monomeric complex is formed through the chelation of a zinc ion by histidine residues H6, H13, and H14 and glutamic acid residue E11 [43]. In addition, H6 can be involved in an inter-peptide coordination of zinc ions by the pairs H6/H13 and/or H14/E11 of two peptide molecules, which suggests a Zn^2+^-bridging mechanism of Aβ oligomerization [45]. The exclusion of the H6 residue from the zinc chelation pattern was shown to block the Zn^2+^-driven Aβ16 oligomerization due to a formation of stable Zn^2+^-mediated dimers via the coordination of Zn^2+^ by the ^11^EVHH^14^ regions of two H6R-Aβ16 peptides [44,45]. However, in the case of full-length peptides, H6R mutation did not preclude their Zn^2+^-induced aggregation (Figure 1 and Figure 2). It is most likely that the lack of histidine residue H6 due to H6R substitution is compensated for by another amino acid residue of Aβ MBD. Indeed, amino acid residues such as D1, E3, and D7 are potentially capable of taking a part in coordinating Zn^2+^ [46]. Obviously, this should result in the structuring of MBD of the mutant peptide upon Zn^2+^ binding, different from that of the intact peptide. Since the structural rearrangement of MBD, caused by Zn^2+^ binding, is known to lead to a conformational reorganization of the C-terminal region in the intact Aβ [47,48], the different structuring of MBD due to the lack of H6 may alter the conformational reorganization of the C-terminal domain and, consequently, the structure of amorphous aggregates.

The double substitution R5A/H6A was reported to have no effect on the Zn^2+^-triggered “coil-to-β-sheet” conformational transition in Aβ40 peptides [15]. We could find no experimental studies devoted to the impact of R5A substitution per se on either the self- or Zn^2+^-induced Aβ aggregation. Nonetheless, the molecular dynamic simulation study [29] demonstrated that such substitution can alter the structural organization of both N- and C-terminal domains of Aβ42. Apparently, the change in Aβ42 structural organization due to the concurrent substitution R5A leads to the different response of R5A/H6A-Aβ40 peptides to Zn^2+^, compared to that of H6R-Aβ42.

## 5. Conclusions

The English (H6R) mutation, while not preventing the Zn^2+^-triggered aggregation of the Aβ42 peptide, was found to influence the conformational alterations in the peptide, induced by its complexing with Zn^2+^, and, consequently, the structural characteristics of Zn^2+^-induced amorphous aggregates. The mutant peptide H6R-Aβ42 showed no appreciable changes in its structure under molar excess of Zn^2+^, in contrast to the intact Aβ42 peptide which demonstrated an increase in the random coil and β-sheet content. The formed Zn^2+^-induced amorphous aggregates of the H6R mutant appeared structurally different from those of the intact Aβ42 that was manifested by their distinct optical properties. The presented findings further advance the understanding of the pathological role of H6R mutation. Furthermore, they highlight the possibility that some other amino acid residue of Aβ MBD may participate in the coordinating Zn^2+^ when H6 is excluded from the chelation pattern. This possibility should be taken into account in the rational design of anti-AD drugs aimed at disrupting pathogenic Zn^2+^-^+^Aβ interactions.

## Figures and Tables

**Figure 1 biomolecules-10-00961-f001:**
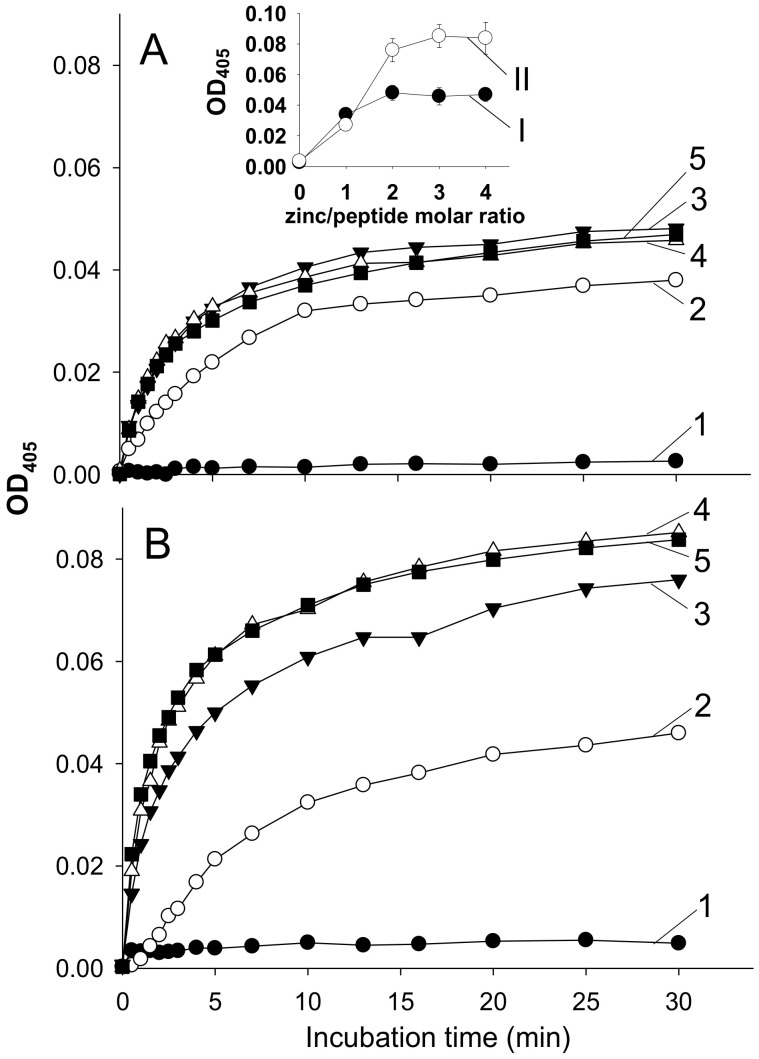
Dependence of turbidity (optical density at 405 nm, OD_405_) of Aβ42 and H6R-Aβ42 preparations on the incubation time after the addition of Zn^2+^ at different Zn^2+^/peptide molar ratios. Panel (**A**)—Aβ42; panel (**B**)—H6R-Aβ42. Curves 1, 2, 3, 4, and 5 correspond to the Zn^2+^/peptide molar ratios of 0, 1, 2, 3, and 4, respectively. Peptides in buffer H (10 mM HEPES, pH 6.8, 50 mM NaCl); peptide concentration—25 µM. Data points are means of three measurements; relative standard deviations for any of the points did not exceed 15%. The insert in panel (**A**) shows the turbidity at the 30th min of incubation as a function of the Zn^2+^/peptide molar ratio. Curves I and II—Aβ42 and H6R-Aβ42 preparations, respectively.

**Figure 2 biomolecules-10-00961-f002:**
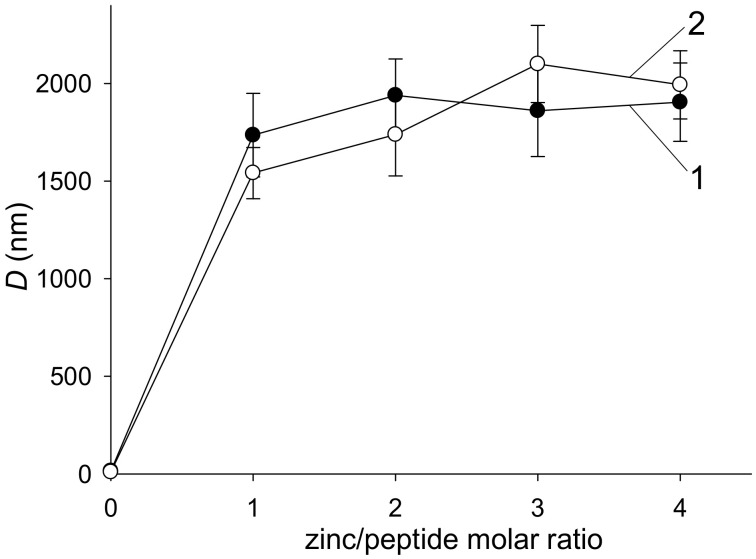
The characteristic diameter (*D*) of Zn^2+^-induced Aβ aggregates, measured after 30-min incubation as a function of the Zn^2+^/peptide molar ratio. Curves 1 and 2—Aβ42 and H6R-Aβ42 preparations, respectively. Peptides in buffer H (10 mM HEPES, pH 6.8, 50 mM NaCl); peptide concentration—25 µM. The means and standard deviations for three measurements are shown.

**Figure 3 biomolecules-10-00961-f003:**
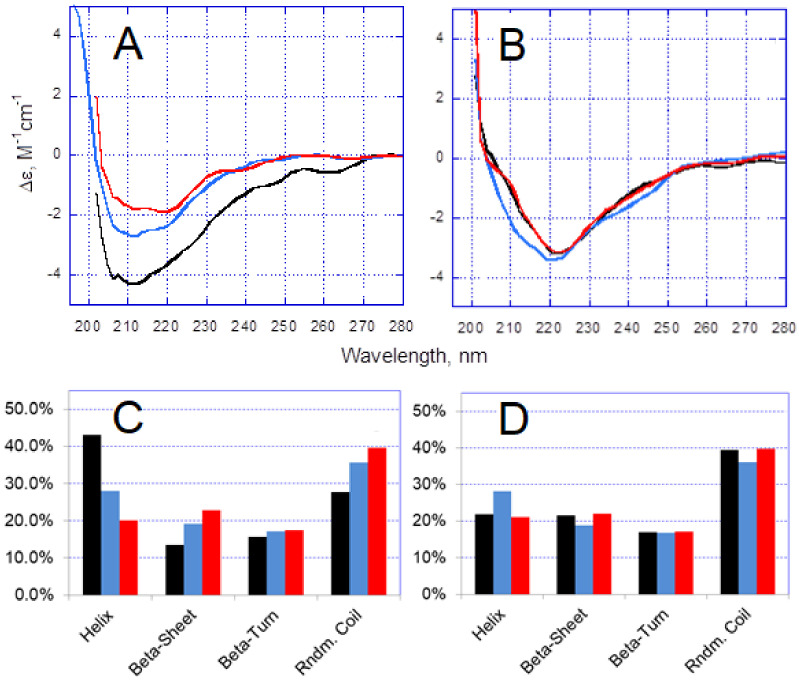
Circular dichroism spectra (**A**,**B**) and fractions of the secondary structure components drawn from the spectra with CDNN software (**C**,**D**). Peptides Aβ42 (**A**,**C**) and H2R-Aβ42 (**B**,**D**). CD spectra were collected in 20 min of incubation either in the absence or in the presence of Zn^2+^. Each CD spectrum is an average of three separate measurements. “Beta-Sheet” is a sum of parallel and antiparallel β-sheet components. The Zn^2+^ concentrations are indicated by color: Black—no Zn^2+^ added, blue and red—50 and 100 µM Zn^2+^, respectively. Peptides in buffer H (10 mM HEPES, pH 6.8, 50 mM NaCl); peptide concentration—25 µM.

**Figure 4 biomolecules-10-00961-f004:**
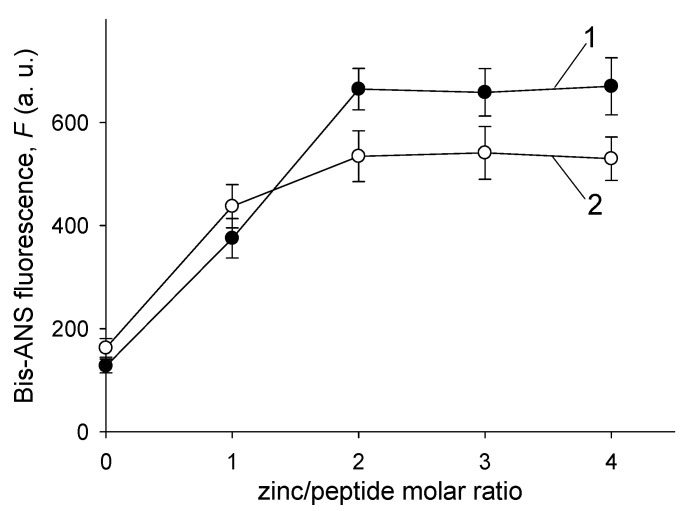
Dependencies of fluorescence of bis-ANS/Aβ complexes, *F*, on the Zn^2+^/peptide molar ratio. The fluorescence was measured 30 min after the addition of Zn^2+^. Curves 1 and 2—Aβ42 and H6R-Aβ42, respectively. Peptides in buffer H (10 mM HEPES, pH 6.8, 50 mM NaCl); peptide concentration—25 µM. Bis-ANS/peptide molar ratio—1:10. The means and standard deviations for triplicate measurements are shown.

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
