# Peer review of "The English (H6R) Mutation of the Alzheimer’s Disease Amyloid-β Peptide Modulates Its Zinc-Induced Aggregation"

_biomolecules, 2020, doi:10.3390/biom10060961_

Round 1

Reviewer 1 Report

The paper  “The English (H6R) mutation of the Alzheimer’s 1 disease amyloid-β peptide modulates its zinc-induced 2 aggregation”  by Radko et al. tries to characterize the conformational properties of the H6R mutation of amyloid-beta peptide in the presence of the Zn ion, using CD, fluorescence and sedimentation techniques.

The major problem of the paper is that the results, either in presence or absence of Zn2+, are the aggregation of the peptide, and using the above techniques it is impossible to distinguish about different forms of aggregation. What is surprising to me is the fact that in a previous paper that include some of these authors (Istaret et al., Sci. Rep., 2016) using NMR and molecular dynamics, they were able to propose a two-steps mechanism of aggregation based on NMR data. In this mechanism an important role is played by residue E11, which is present both in the dimer and in the oligomer, whilst in the present paper the role of E11 is neglected. Anyhow, the mechanism presented in that paper (Fig. 8) perfectly explains the role of H6 that, when mutated, may alter (but not prevent) the aggregation mechanism.

In conclusion, the experimental evidences presented in this paper are not conclusive from the conformational point of view and in my opinion they do not add enough new information.

Author Response

The paper “The English (H6R) mutation of the Alzheimer’s 1 disease amyloid-β peptide modulates its zinc-induced 2 aggregation” by Radko et al. tries to characterize the conformational properties of the H6R mutation of amyloid-beta peptide in the presence of the Zn ion, using CD, fluorescence and sedimentation techniques.

The major problem of the paper is that the results, either in presence or absence of Zn2+, are the aggregation of the peptide, and using the above techniques it is impossible to distinguish about different forms of aggregation. What is surprising to me is the fact that in a previous paper that include some of these authors (Istaret et al., Sci. Rep., 2016) using NMR and molecular dynamics, they were able to propose a two-steps mechanism of aggregation based on NMR data. In this mechanism an important role is played by residue E11, which is present both in the dimer and in the oligomer, whilst in the present paper the role of E11 is neglected. Anyhow, the mechanism presented in that paper (Fig. 8) perfectly explains the role of H6 that, when mutated, may alter (but not prevent) the aggregation mechanism.

In conclusion, the experimental evidences presented in this paper are not conclusive from the conformational point of view and in my opinion they do not add enough new information.

The techniques employed in our study included the thioflavin T fluorescence assay that allowed us to distinguish fibrillary and amorphous aggregates. Our results show that, for both peptide isoforms under conditions used, the formed zinc-induced aggregates are amorphous and, therefore, they lack any defined regular structural organization in contrast to fibrillar Aβ aggregates. The amorphous Aβ aggregates are rather assemblies of randomly packed Aβ peptides with an irregularly distribution (over the aggregate volume) of the inclusions of ordered sites of polypeptide chains (alpha-helixes, beta-sheets) due to intra- and, perhaps, interpeptide interactions. Thus, though each individual peptide can have some structured regions, the distribution of such regions in the aggregate per se is irregular. To decipher a precise structure of such amorphous aggregates is a quite challenging task, at least. Some integral structural characteristics such as a fraction of particular secondary structure can, at best, be obtained for these aggregates what we did by employing CD spectroscopy.

The present knowledge regarding effects of the H6R substitution (the English familiar mutation associated with the early onset of Alzheimer’s disease) on the zinc-induced Aβ aggregation implicated in Alzheimer’s disease is contradictive. On one hand, the H6A substitution in Aβ28 and Aβ42 peptides was shown not to prevent the zinc-induced Aβ aggregation in vitro (refs. 15 and 16 of the revised manuscript) but, on the other hand, the H6R substitution in Aβ16 peptides had a dramatic suppressive effect on the zinc-triggered peptide oligomerization die to the formation of stable Aβ16-Zn2+-Aβ16 dimers (refs. 44 and 45 of the revised manuscript; ref. 45 is the paper by Istrate et al., Sci. Rep., 2016, mentioned by the Reviewer above). As we stated in the Introduction, “the aim of the present work was to get further insights into the effects of H6R substitution on the in vitro aggregation of the Aβ42 peptide, triggered by Zn2+” (lines 68-69 of the revised manuscript). The Aβ16 peptides, both in the intact and mutated forms, are a well-established model for deciphering the structure of zinc-Aβ complexes but a poor model for studying the zinc-induced Aβ aggregation due to a lack of the hydrophobic C-terminal domain. In the frame of this study we did not set a particular aim of deciphering a precise conformational structure of aggregates induced by zinc ions for peptide isoforms studied since it is hardly possible.

Nonetheless, the amorphous aggregates may still be structurally different in terms of, e.g., density of peptide random packing (we specifically mentioned such possibility upon revision; lines 261-262). Obviously, such difference would be tightly connected to structural organization of individual peptides in aggregates. Thus, we have first focused on the integral (macroscopic) characteristics of these aggregates, employing the appropriate techniques such as turbidimetry and dynamic light scattering. The obtained data clearly indicated that the macroscopic characteristics of the formed aggregates are quite different for the intact and mutated Aβ42 peptides: their optical properties as light scattering centers are distinct. This observation was supported by the results of the bis-ANS fluorescence assay that together allowed us to conclude that aggregates of the intact and mutated peptides are structurally dissimilar. The dissimilarity was ascribed to different conformational alterations triggered by the zinc binding to these peptides, as revealed by CD spectroscopy. These are original results giving a new view of the impact of the H6R mutation on the zinc-induced aggregation of the full-length Aβ peptide.

Finally, we would like to point that the residue E11 cannot play any crucial role in the studied case since it is present in both the intact and mutated peptide.

Reviewer 2 Report

This is an excellent piece of work, using a range of spectroscopic and analytical techniques to probe the effect of a known mutation of amyloid-beta on the zinc-induced aggregation. The experimental design is excellent, as are the reported data and its presentation. A clear signal is found, and likely reasons for this effect are put forward.

I have no hesitation in recommending it for your journal, and have no suggestions for improvement.

Author Response

This is an excellent piece of work, using a range of spectroscopic and analytical techniques to probe the effect of a known mutation of amyloid-beta on the zinc-induced aggregation. The experimental design is excellent, as are the reported data and its presentation. A clear signal is found, and likely reasons for this effect are put forward.

I have no hesitation in recommending it for your journal, and have no suggestions for improvement.

We are thankful to the Reviewer for an appreciation of our work.

Reviewer 3 Report

The present manuscript focuses on the investigation of H6R substitution on Zn-induced aggregation pathway of AB42. As a result, the mutation was found to lead to different conformation of the protein assembly. This is an interesting study for the readership of Biomolecules. I ask the authors to discuss the following points:

1) the authors say that amyloid plaques are abnormally loaded with zinc. However, I would ask the authors to specify if the Zn concentration used in this study is in line with the physiological value, and if not, how this would impact the obtained results.

2) The authors have estimated the amount of secondary structure of the AB assemblies. I ask the authors to include a clear comparative analysis between their results and the data available in literature about the AB content of secondary structures.

3) Several studies in literature have discussed the role of metallic ions (10.1007/s12274‐017‐1734‐9) and protein mutations (10.1038/nature01891, 10.1080/07391102.2019.1671224) on amyloid beta aggregation mechanisms. The authors should discuss their results also in the context of the literature in this field.

Author Response

The present manuscript focuses on the investigation of H6R substitution on Zn-induced aggregation pathway of AB42. As a result, the mutation was found to lead to different conformation of the protein assembly. This is an interesting study for the readership of Biomolecules. I ask the authors to discuss the following points:

1) the authors say that amyloid plaques are abnormally loaded with zinc. However, I would ask the authors to specify if the Zn concentration used in this study is in line with the physiological value, and if not, how this would impact the obtained results.

The zinc concentration in our study ranged from zero up to 200 µM. Zinc ions are known to be transiently present in synaptic clefts at concentrations up to 300 µM when released from synaptic vesicles during neuronal excitation (e.g., ref. 10 of the revised manuscripts). Thus, the zinc concentration used in the study is in line with the physiological value.

2) The authors have estimated the amount of secondary structure of the AB assemblies. I ask the authors to include a clear comparative analysis between their results and the data available in literature about the AB content of secondary structures.

Though there are numerous reports on zinc-triggered conformational transitions in Aβ peptides, monitored with CD spectroscopy (e.g., PMID: 11054124; 16266835; 16672274; 19539421; 19083027; 21216965; 23844690; 24119373; 29476859), the majority of them analyze data in a qualitative way, simply as changes in CD spectra. The exceptions are the works by Drochioiu et al. (PMID: 19539421) and Chen et al. (PMID: 16672274). However, the former provides estimates for the secondary structure content merely in the absence of zinc while the later deals with zinc-induced Aβ assemblies formed in the presence of trifluoroethanol as an additive to aqueous buffer. Nonetheless, there is a general consent that zinc ions trigger a transition to Aβ conformations with a higher β-sheet content. To address the reviewer recommendation, we added the two most relevant references (the full-length Aβ42 peptide, no trifluoroethanol) to point at this fact (refs. 32 and 33, line 182). Also, we discussed the paper of Janaszewska et al., suggested by the Reviewer below (doi: 10.1007/s12274‐017‐1734‐9) where authors provide a comprehensive secondary structure estimate for the case of another divalent metal, copper, known to trigger a rapid Aβ aggregation (ref. 34, lines 181-185 of the revised manuscript).

3) Several studies in literature have discussed the role of metallic ions (10.1007/s12274‐017‐1734‐9) and protein mutations (10.1038/nature01891, 10.1080/07391102.2019.1671224) on amyloid beta aggregation mechanisms. The authors should discuss their results also in the context of the literature in this field.

The suggested studies were cited and discussed in the revised manuscript (ref. 34, line 185; ref. 41, line 249; ref. 42, line 250).

Round 2

Reviewer 1 Report

See below in the comments to Editors.